# Aging Induces Profound Changes in sncRNA in Rat Sperm and These Changes Are Modified by Perinatal Exposure to Environmental Flame Retardant

**DOI:** 10.3390/ijms21218252

**Published:** 2020-11-04

**Authors:** Alexander Suvorov, J. Richard Pilsner, Vladimir Naumov, Victoria Shtratnikova, Anna Zheludkevich, Evgeny Gerasimov, Maria Logacheva, Oleg Sergeyev

**Affiliations:** 1Department of Environmental Health Sciences, School of Public Health and Health Sciences, University of Massachusetts, 686 North Pleasant Street, Amherst, MA 01003, USA; rpilsner@umass.edu; 2Belozersky Institute of Physico-Chemical Biology, Lomonosov Moscow State University, Leninskye Gory, House 1, Building 40, 119992 Moscow, Russia; vtosha@yandex.ru (V.S.); maria.log@gmail.com (M.L.); olegsergeyev1@yandex.ru (O.S.); 3Bioinformatics Laboratory, Kulakov National Medical Research Center of Obstetrics, Gynecology and Perinatology Ministry of Health of the Russian Federation, Oparina 4, 117997 Moscow, Russia; looongdog@gmail.com; 4Genomed Ltd., Baumanskaya 50/12, Bld. 1, 105005 Moscow, Russia; annabeth495@gmail.com; 5E.I. Martsinovsky Institute of Medical Parasitology and Tropical Medicine, I.M. Sechenov First Moscow State Medical University, 20 Malaya Pirogovskaya, 119435 Moscow, Russia; jalgard@gmail.com; 6Faculty of Biology, Lomonosov Moscow State University, 119992 Moscow, Russia; 7Center for Life Sciences, Skolkovo Institute of Science and Technology, 143028 Moscow, Russia; 8Chapaevsk Medical Association, Meditsinskaya Str. 3a, Samara Region, 446100 Chapaevsk, Russia

**Keywords:** aging, paternal exposure, sperm, semen, epigenetics, sncRNA, piRNA, miRNA, 2,2′,4,4′-tetrabromodiphenyl ether, PBDE, BDE-47, perinatal, environment

## Abstract

Advanced paternal age at fertilization is a risk factor for multiple disorders in offspring and may be linked to age-related epigenetic changes in the father’s sperm. An understanding of aging-related epigenetic changes in sperm and environmental factors that modify such changes is needed. Here, we characterize changes in sperm small non-coding RNA (sncRNA) between young pubertal and mature rats. We also analyze the modification of these changes by exposure to environmental xenobiotic 2,2′,4,4′-tetrabromodiphenyl ether (BDE-47). sncRNA libraries prepared from epididymal spermatozoa were sequenced and analyzed using DESeq 2. The distribution of small RNA fractions changed with age, with fractions mapping to rRNA and lncRNA decreasing and fractions mapping to tRNA and miRNA increasing. In total, 249 miRNA, 908 piRNA and 227 tRNA-derived RNA were differentially expressed (twofold change, false discovery rate (FDR) *p* ≤ 0.05) between age groups in control animals. Differentially expressed miRNA and piRNA were enriched for protein-coding targets involved in development and metabolism, while piRNA were enriched for long terminal repeat (LTR) targets. BDE-47 accelerated age-dependent changes in sncRNA in younger animals, decelerated these changes in older animals and increased the variance in expression of all sncRNA. Our results indicate that the natural aging process has profound effects on sperm sncRNA profiles and this effect may be modified by environmental exposure.

## 1. Introduction

There is a growing trend of delayed parenthood in developed countries as a consequence of increased life expectancy, socioeconomic pressures, and rates of divorce and remarriage. For example, in the USA, a 50% increase over the past 30 years has been reported in the number of men fathering children in the age range of 35–44 years [1] and the mean paternal age has increased over the past 44 years from 27.4 to 30.9 years [2]. Although self-renewal and differentiation of spermatogonial stem cells permits the production of mature sperm throughout the adult life course, this benefit of continued spermatozoa production may also come at the expense of the accumulation of genetic and epigenetic errors over the life course, which may, in turn, have downstream consequences for the health and development of offspring.

Epidemiologic evidence demonstrates that advanced paternal age is associated with a host of adverse offspring health outcomes. The adverse conditions associated with increased paternal age at conception include stillbirths [3,4], musculoskeletal syndromes [5], cleft palate [6], acute lymphoblastic leukemia [7,8], retinoblastoma [9,10], and polygenic neurodevelopmental and psychiatric disorders [11], such as schizophrenia [12,13], autism spectrum disorders [14,15], bipolar disorder [16] and attention deficit/hyperactivity disorder [17]; however, the causal link between paternal age and offspring health is not yet well understood.

Spermatozoa have been traditionally deemed vehicles for the transfer of the paternal haploid genome upon fertilization; however, epigenetic marks on sperm (e.g., DNA methylation, histone modifications and small noncoding RNA) can act as a legacy of environmental exposures encountered throughout the life course that affect the early-life development and offspring phenotype [18,19]. To that end, it has been hypothesized that an altered offspring phenotype is associated with age-related accumulation of epigenetic changes in the sperm of fathers [20,21].

Increasing evidence indicates that sperm small non-coding RNA (sncRNA) is an important vehicle connecting paternal experiences with phenotypes of their offspring [19,22,23]. The most elaborate research connects paternal diet, the composition of sperm sncRNA and offspring development. For example, protein restriction in mice affected sncRNA levels in sperm, including decreased let-7 miRNA and altered composition of tsRNA [24]. Specifically, in that study, changes in the sperm load of tRF-Gly-GCC were associated with endogenous retroelement MERVL expression in early embryos. In mouse experiments, a high-fat diet changed the expression of X-linked miRNA [25] and tRNA-derived small RNAs (tsRNA) [26]. Injection of these tsRNAs into normal zygotes altered metabolic pathways in embryos [26]. Similarly, the injection of sperm RNA from mice fed a Western-like diet into naive one-cell embryos produced changes in the metabolic phenotype of the resulting progenies normally induced by a Western-like diet [27].

High-fat diet-induced obesity in mice was associated with altered expression of around 400 miRNA [28]. Interestingly, in this study, paternal phenotype characterized by obesity and insulin resistance was transferred in two consecutive generations of both sexes’ offspring. In another mouse study, high-fat diet altered the sperm levels of 15 miRNA, 41 tsRNA and 1092 piRNA [29]. Contrary, to previous research, F1 and F2 female offspring of treated fathers were resistant to high-fat diets. The connection between obesity and the sperm profile of sncRNA is supported by human research as well. For example, in one study, lean and obese men had significantly different levels of 37 piRNAs in their spermatozoa [30].

The ability of chemical exposure to affect the composition of sncRNA in sperm only has started to emerge. For example, in mice, chronic ethanol exposure altered several sncRNAs, including tsRNAs, mitochondrial small RNA, and miRNA [31]. The same study also reports that tsRNAs were similarly altered in sperm and epididymosomes, supporting the hypothesis that altered tsRNAs are loaded to spermatozoa in the epididymis [32]. To the best of our knowledge, the effects of paternal exposures to environmental xenobiotics on the composition of sperm sncRNA have not yet been addressed by existing research. Similarly, there is a knowledge gap in the understanding of the role of paternal age in the sperm load of sncRNA.

Given the general trend of delayed parenthood in developed countries, it is important to clarify the effect of paternal aging on the sperm epigenome, as well as the impact of environmental exposure on age-related changes in the sperm epigenome. In the current study, we first examine the effect of aging on the sperm small RNA by comparing sncRNA profiles in control rats on postnatal day (PND)65 and PND120, approximately corresponding to young pubertal and mature men, respectively [33,34]. Furthermore, we analyze how observed age-dependent changes in sncRNA are modified by perinatal exposure to a ubiquitous environmental flame retardant, 2,2′,4,4′-tetrabromodiphenyl ether (BDE-47), in doses relevant to background human exposure. Our results indicate that the natural aging process has profound effects on the sperm profiles of sncRNA and these effects may be modified by environmental exposure.

## 2. Results

In this study we analyzed the effects of age and exposure to a ubiquitous environmental flame retardant, 2,2′,4,4′-tetrabromodiphenyl ether (BDE-47), on sncRNA profiles in rat sperm. Dams were exposed orally between pregnancy day 8 and postnatal day 21 (PND21) and sncRNA were analyzed in the caudal epididymis sperm of offspring on PND65 and PND120 using a next-generation sequencing approach (Figure 1). We found no significant relationship between litter size and exposure group, with the number of pups varying from 10 to 15 per litter. Litter size was 12.33 ± 0.51 in the control group and 12.71 ± 0.56 in the BDE-47-exposed group (all data are for mean ± SE). No weight differences were observed between the control and exposed dams or pups throughout the experiment.

### 2.1. Changes in the Profile of Small RNA Expression

Sequencing was completed with an average 16.5 million reads per sample with a range of 6.5–23.3 million reads and 80% average alignment to the reference genome. Sequenced fragments were distributed in a range between 16 and 46 nucleotides. The distribution of different types of non-coding RNA, as analyzed using RNAcentral, shows high consistency across samples within age groups (Figure 2A). The distribution of sncRNA subtypes was different between age groups (chi-square *p* = 8.91 × 10^−13^ for controls and 3.70 × 10^−5^ for BDE-47-exposed groups). Age-related changes were similar for both exposed and control groups (Figure 2A). Percent reads mapping to rRNA and lncRNA decreased with age from 46.2 ± 2.5% to 37.1 ± 2.3% for rRNA and from 24.4 ± 1.3% to 16.3 ± 2.0% for lncRNA (mean ± SE). Percent reads mapping to tRNA, precursor miRNA and miRNA increased with age from 9.8 ± 1.5% to 16.5 ± 1.5% for tRNA, from 2.4 ± 0.4% to 9.4 ± 1.8% for precursor miRNA and from 0.6 ± 0.1% to 3.2 ± 0.7% for miRNA. Percent reads of piRNA did not change significantly with age (16.6 ± 1.6% on PND65 and 17.7 ± 1.5% on PND120). The distribution of reads mapping to different RNA types was not significantly different between exposure groups of each age (chi-square *p* = 0.68 for PND65 and 0.69 for PND120).

A principle component analysis of all 24 biological samples based on miRNA expression shows a clear grouping of control animals by age (Figure 2B). Samples collected from exposed animals do not show clear groupings. Similarly, control samples cluster by age in a heatmap of the top 100 miRNAs with the highest standard deviations, while exposed samples do not form clear age-related clusters (Figure 2C).

Values of differential expression were identified for 1038 individual miRNAs and precursor miRNAs, 37,459 piRNA and 444 tRNA (Appendix A) using default independent filtering and Coock’s cut-off parameters in the DESeq2 package [36]. Given that the distribution of RNA sizes in our study does not cover precursor miRNA and tRNA, identification of these molecules is likely due to the presence of their fragments in our samples or due to multiple mappings of shorter RNA, such as the mapping of miRNA to precursor miRNA. Numbers of individual sncRNA significantly differentially expressed (fold change ≥2 or ≤−2 and false discovery rate (FDR) adjusted *p* ≤ 0.05) in age and exposure groups are shown in Table 1. Age has the highest effect on non-coding RNA expression (1.384 differential expressed sncRNA in control group) while exposure only has a minor effect (1 and 24 differentially expressed sncRNA on PND65 and 120, respectively). Interestingly, exposure seemingly attenuates the effect of age on RNA expression (1.384 in control group vs. 165 in exposed group).

We hypothesized that the later observation may be at least partly due to the fact that exposure increases variance in expression levels across biological replicates; therefore, a smaller number of individual RNAs passes our threshold of FDR-adjusted *p*-value. To test this hypothesis, we compared the variance in expression values across age and exposure groups (Figure 2D). This comparison indeed shows that exposure increases the variance of non-coding RNA in both age groups. It is difficult to infer whether increased variance is the only factor responsible for the decreased number of significantly differentially expressed age-dependent RNA. To test if additionally exposure attenuates the effect of age we compared, across all groups, the expression values of age-dependent sncRNAs significantly altered between two age groups in control animals (Appendix A).

According to the distribution of expression values, exposure to BDE-47 produces changes concordant with accelerated aging in younger animals and attenuates the effect of age in older animals. Specifically, the expression of miRNA and piRNA undergoing age-dependent suppression in control rats is lower in exposed animals on PND65 than in controls of the same age (Appendix A). Likewise, piRNA undergoing age-dependent increases in expression have higher levels of expression in young exposed animals than in young control animals (Appendix A). Thus, exposure-induced changes in the expression of these groups of sncRNA in young rats resemble age-dependent changes in control animals. Similarly, exposure attenuates age-dependent changes in some groups of sncRNA. In particular, miRNA and piRNA undergoing age-dependent increases in expression have lower levels of expression in older exposed animals as compared with control rats (Appendix A). The same trend is seen for tRNAs (Appendix A).

Among significant age-dependent sncRNA, 55 miRNA, 39 tRNA and 5 piRNA overlapped between control and exposed groups (Figure 2E–G). Out of the 55 overlapping miRNA, 52 underwent age-dependent changes in the same direction in both exposure groups and, for three miRNA, the direction of change was opposite in control and exposed groups. All overlapping tRNA and piRNA had the same direction of age-dependent change in control and exposed groups. Pearson’s correlation of Log2 fold change values in exposed and control animals was 0.61 for the list of overlapping miRNA, and 0.96 for overlapping tRNA or piRNA (Appendix A). To check if, in both exposed and control animals, age-dependent changes occur in the same direction, we also calculated Pearson’s correlation of Log2 fold change values for merged lists of sncRNA that were significantly differentially expressed in either exposure group or both. Pearson’s correlation was 0.60 for the merged list of 262 miRNA, 0.78 for the merged list of 241 tRNA, and 0.52 for the merged list of 947 piRNA (Appendix A).

Exposure to BDE-47 altered the expression of only one miRNA on PND65 (Appendix A), which was upregulated 18-fold in exposed animals. On PND120, 18 miRNA were significantly upregulated in response to exposure (Appendix A). Changes in the expression of these RNAs on PND65 and 120 were not consistent (Appendix A). Six tRNA were significantly differentially expressed in response to BDE-47 on PND120. However, no tRNA passed a threshold of significance on PND65, although the expression of four tRNA out of six differentially expressed on PND120 were changed in the same direction (Appendix A). No piRNA passed the threshold of significance at either time point.

Interestingly, the majority of sncRNA altered in response to exposure (one out of one miRNA changed on PND65, 16 out of 18 miRNA changed on PND120 and six out of six tRNA changed on PND120) were in the merged list of significant age-dependent RNA, i.e., they were differentially expressed between PND65 and 120 in control or exposed groups or both.

### 2.2. Functional Analysis of Age-Dependent Changes in miRNA Expression

We first uploaded to the miRDP database—an online database for predicted microRNA targets in animals [37,38]—the list of 249 age-dependent miRNAs differentially expressed in control animals, and identified 4908 genes—targets of miRNA with prediction scores ≥ 80. Many identified genes were targets of several differentially expressed miRNA (Appendix A). Given that Metascape can analyze only datasets not exceeding 3000 genes, we uploaded to this tool a list of 2654 genes, targets of at least two miRNA. This list was highly enriched with high statistical significance (−log 10(P) ranged 14–23) for a broad range of developmental categories (Figure 3A). To add detail to this analysis, we further restricted it to gene targets of ≥five miRNA undergoing age-dependent increases or decreases in expression.

Targets of upregulated miRNA were enriched for different categories related to embryonic and other types of development (ex.: axon development, appendage development, sensory organ development, odontogenesis), apoptosis and the cell cycle (ex.: cell death signaling via NRAGE, NRIF and NADE, positive regulation of cell cycle, leucocyte apoptotoic process), lipid metabolism (ex.: NR1H2 and NR1H3-mediated signaling, regulation of lipid metabolic process) and oxidative stress (response to redox state) (Appendix A). Targets of downregulated miRNA were also enriched for developmental categories (ex.: embryonic organ development, response to growth factor, reelin pathway), oxidative stress (AGE-RAGE signaling pathway in diabetic complications), and the cell cycle (positive regulation of mitotic cell cycle), as well as for the metabolism of carbohydrates (ex.: glucose transmembrane transport, response to carbohydrate) and other types of metabolism regulation (ex.: positive regulation of cold-induced thermogenesis, fat cell differentiation) and transcriptional and posttranscriptional regulation (ex.: poly(A) + mRNA export from nucleus, transcriptional regulation of pluripotent stem cells, production of miRNA involved in gene silencing in miRNA) (Appendix A).

The majority of differentially expressed miRNA between PND65 and PND120 in exposed animals overlapped with age-dependent miRNA in control animals (Figure 2E). Therefore, in the next step, we focused on 55 significantly differentially expressed miRNA between PND65 and PND120 in both control and BDE-47-exposed animals. Using the miRDP database, we identified 2261 gene targets of these 55 miRNA with prediction scores ≥ 80. The use of the entire list of these gene targets for Metascape analysis showed high enrichment with high statistical significance (−log 10(P) ranged 12–27) of mostly the same developmental categories as those affected by age in control animals (Figure 3B,C).

### 2.3. BDE-47 Exposure-Dependent Changes

Neurotransmitter receptor transfer to a plasma membrane was the top biological category enriched for 106 targets of the one miRNA differentially expressed on PND65 in response to BDE-47 exposure (Appendix A). We then identified 688 gene targets of 18 differentially expressed miRNA on PND120 in response to BDE-47 exposure. The top biological category enriched by these genes was the modulation of chemical synaptic transmission (Figure 3D). Additionally enriched biological categories include pathways of growth and development (ex.: regulation of growth, developmental growth, regulation of bone resorption, cell part morphogenesis), the regulation of carbohydrate metabolism (ex.: regulation of glucose transmembrane transport, phosphatidylinositol signaling system, insulin resistance) and others. However, it should be noted that the significance of BDE-47 exposure-dependent changes are smaller (−log 10(P) range: 2–6) than for age-dependent changes.

### 2.4. Functional Analysis of Age-Dependent Changes in piRNA Expression

Using data on rat piRNA functional annotation from piRBase [39], we analyzed whether age-dependent differentially expressed piRNAs in sperm are enriched with piRNA targeting different genomic elements. Our null hypothesis assumed that age and expression in the sperm of piRNA targeting specific genomic elements are independent variables. Compared with numbers predicted from the null hypothesis, we found that, in sperm of control animals, piRNA targeting transposable elements, long interspersed elements (LINE) and short interspersed elements (SINE) as well as satellite DNA sequences were not significantly over- or under-represented among age-dependent piRNA (Table 2).

piRNA targeting long terminal repeats (LTRs) and protein-coding genes were significantly (*p* < 0.00001) overrepresented in the sperm of control animals among age-dependent piRNA with 5.3- and 11.8-fold enrichment, respectively (Table 2). In exposed animals, enrichment of age-dependent piRNA for different genomic elements was significant only for LTR (*p* = 0.026, enrichment fold change = 3.5).

Lastly, in the spermatozoa of control animals, we identified 142 age-dependent piRNA targeting protein-coding genes. These piRNAs were involved in 158 piRNA–gene pairs (some piRNA targeting more than one gene) with 42 genes (some genes targeted by more than one piRNA). Among these genes, Ppil1 was targeted by 88 piRNA, Add2 by 14, Stox1 by 9, Loc500567 by 5, Brinp2 by 3 and Loc360933 and Cep830s by 2 piRNA. All other genes were targets of one piRNA. Metascape analysis of the list of 42 genes—targets of age-dependent piRNA—showed the enrichment of metabolic and developmental processes (Figure 3E). In exposed animals, enrichment of age-dependent piRNA for different genomic elements was significant only for LTR (*p* = 0.026, enrichment fold change = 3.5) (Appendix A).

## 3. Discussion

In this study, we report, for the first time, that the composition of sncRNAs in rat sperm undergoes age-dependent changes, where fractions of rRNA-derived sncRNA and lncRNA decreased with age, while fractions of tRNA-derived sncRNA and miRNA increased with age and fractions of piRNA did not change with age. These findings may have fundamental importance for the understanding of the mechanisms involved in age-dependent changes in epigenetic information transferred by spermatozoa, as well as age-dependent changes in sperm quality and fertility. In our study, more than 1000 sncRNAs, including miRNA, tRNA and piRNA underwent significant age-dependent changes in their expression in rat spermatozoa, which were enriched in the regulation of genes involved in developmental and metabolic processes (Figure 3). Finally, perinatal exposure of rats to low dose of environmental flame retardant modifies these age-dependent changes in sncRNA by accelerating age-dependent changes in younger animals and attenuating them in older animals.

### 3.1. sncRNA in Sperm

The role of sncRNA in spermatogenesis was reviewed in a recent paper [40]. The composition of different sncRNAs changes dynamically throughout the spermatogenesis cycle (74 days in humans, 56 days in rat [22,24,41]. piRNA are mostly germline-specific sncRNA, which have two waves of expression in spermatogenesis. One population of piRNA is upregulated in primordial germ cells and another in pachytene spermatocytes [42,43]. The major recognized role of piRNA consists of the suppression of transposable elements during epigenetic reprograming events [44]. Concordant with this role, piRNAs of the first wave mostly target transposable elements, while piRNAs of the second wave are enriched in protein-coding mRNA targets [45]. In mature human spermatozoa, piRNA expression decreases and miRNA and small RNA derived from tRNA and rRNA are present in much higher quantities. In fact, the two latter types of RNA constitute about 75% of all sncRNA in humans [30,46].

After leaving the testis, mature spermatozoa receive additional loads of miRNA and fragments of tRNA and rRNA delivered by epididimosomes [47] and prostasomes [48]. Mouse models suggest that the load of these extracellular vesicles have significant functional roles and may affect sperm competition, fertilization, embryo development and intergenerational inheritance [24,49,50]. The described dynamic and dramatic changes in sncRNA content during spermatogenesis result in highly selective sncRNA loads of mature spermatozoa. Although emerging evidence indicates that sncRNA in spermatozoa may serve as an important channel of epigenetic information transfer to the next generation, our knowledge of factors that affect sperm sncRNA profiles is only starting to emerge. For examples, existing research identified that high-fat diets and low-protein diets can alter the expression of miRNA, piRNA, and tRNA-derived RNA in spermatozoa (reviewed in [51]). Diet-induced changes in the sncRNA profile were associated with phenotypes displaying insulin resistance, altered body weight, and glucose intolerance. To the best of our knowledge, a knowledge gap still exists in the understanding of age-dependent changes in sperm sncRNA. Additionally, the ability of environmental factors to affect age-dependent changes in sperm sncRNA was not previously reported.

### 3.2. Justification of the Model

We analyzed age-dependent changes in sperm sncRNA by comparing the sperm of young pubertal (PND65) and mature rats (PND120). The male reproductive system undergoes significant changes between these time points. Sexual maturation in rats occurs between 41 and 54 days of age when growth hormone pulse amplitudes increase twofold [52]. According to Robb et al. [33], spermatozoa are first registered in the testis by PND45 and in the epididymis tail by PND50 in Wistar rats. Sperm production increases until PND75 and testis weight increases until PND100. Blood testosterone starts to increase on PND40–45, reaches its maximum by PND76 [34], and then decreases gradually until reaching its adult level by PND97. PND55 male rats are also less successful in the insemination of female rats than 90–95-day old male rats, as indicated by number of pregnancies [53]. Thus, the two age groups in our study (PND65 and PND120) represent distinct stages of reproductive maturation in rats, which correspond to young pubertal and mid-life periods in humans.

To analyze the effect of environmental exposure on age-dependent changes in sperm sncRNA, we tried to simulate the exposure of the general population to brominated flame retardant BDE-47, the most prevalent congener of PBDE in human tissues. The median PBDE concentration in adipose tissue from New York urban population is 399 ng/g lipids [54]. In our previous study, exposure of pregnant rats to 0.2 mg/kg body weight of BDE-47 (the same dose used in the current study) resulted in the accumulation of 234.3 ng BDE-47/g lipid in the adipose tissue of dams [55]. Additionally, our dosing paradigm was designed to simulate human exposure dynamics over different life stages. In the general population, exposure to lipophilic brominated flame retardants is highest during the perinatal period of development. Human studies and animal experiments suggest that these compounds accumulate in maternal adipose tissue, mobilize during pregnancy and lactation, and are delivered via cord blood and breast milk to the developing organism [56,57,58]. BDE-47 easily crosses the placenta [59], and is found in the majority of fetal samples in North America [60,61]. Toddlers are exposed to higher doses of brominated flame retardants than adults because of higher rates of dust ingestion [62] and higher rates of food intake [57]. To mimic this exposure scenario, we exposed males perinatally only; thus, all exposures occurred via cord blood and breast milk. This period covers prospermatogonia development, recognized as a sensitive window of epigenetic reprograming [18,63]. During this period, primordial germ cells experience comprehensive losses of methylation (around E13.5) and the establishment of de novo methylation (around E16). The period also includes drastic changes in histone modification [18,64].

### 3.3. Age-Dependent Changes in sncRNA Fraction

We observed age-dependent changes in the composition of different sncRNA types in rat sperm. The fact that many sperm parameters are changing with age is well recognized [65]. Additionally, a plethora of offspring health effects is associated with paternal age, indicating that age is an important factor that determines what information can be transferred to the next generation via spermatozoa. The mechanisms involved in age-dependent changes in spermatozoa physiology as well as changes in information transferred to the next generation are not yet well understood. Our study demonstrates that the composition of sncRNA may be one potential candidate molecular mechanism involved in these changes. This hypothesis is supported, for example, by findings that the smaller content of RNA derived from rRNA is associated with low-quality embryos after in vitro fertilization [46,66].

Additionally, in relation to changes in sncRNA fractions, we analyzed the functional enrichment of differentially expressed miRNA and piRNA between the two age groups, corresponding approximately to young pubertal and mature men. Targets of differentially expressed miRNA were highly enriched with developmental (including neurodevelopment) and metabolic categories. Similarly, protein-coding genes, targets of differentially expressed piRNA, were also enriched for metabolic and developmental processes. Differentially expressed piRNA were also highly enriched for LTR targets. Interestingly, the rat genome contains approximately 556,000 copies of LTR elements, representing 9% of the whole genome [67]. LTRs, also referred to as endogenous retroviruses, are traditionally viewed as threats to genomic stability [68,69]. To avoid exponential amplification of transposable elements (TE), mammalian organisms have developed mechanisms of TE silencing, with the piRNA pathway being the core mechanism of genome protection from TE in a germline. In spermatogenesis, piRNA ensure genome integrity and male fertility [70]. Emerging evidence demonstrates, however, that over a course of symbiotic evolution, these initial parasitic sequences have developed important functions such as increasing the fitness of the host genome [71,72]. Specifically, TEs play a critical role during the early stages of embryo development. Cleavage-stage embryos undergo global epigenetic reprograming and thus provide an environment suitable for the transcription of LTRs [73]. The activation of LTRs during early embryonic development is critical for zygotic genome activation [71]—a critical event in preimplantation embryo development, when two-cell-stage embryos switch from maternal control to zygotic genome control. Many genes activated at this step contain promoters derived from LTR [74,75]. For example, the expression of murine LTR MERVL peak during two-cell embryos [71], and the downregulation of MERVL through RNA interference results in developmental arrest at that stage [76].

Interestingly, other studies from our group, using the same rat model to analyze age- and exposure-associated changes in sperm DNA methylation, reported that, in control animals, genes associated with differentially methylated regions (DMRs) were significantly enriched with gene targets of age-dependent miRNA (*p* < 0.00001) and piRNA (*p* = 0.035) [35]. Additionally, genes (*n* = 1052) overlapping as miRNA targets and as DMR-associated genes were highly enriched (−log 10(P) ranged 14–22) for categories related to embryonic development.

Thus, our results present more than one line of evidence indicating that age-dependent changes in the sperm epigenome target molecular mechanisms involved in basic developmental processes. Such results are in line with epidemiologically and clinically assisted reproduction outcomes, indicating that increased paternal age is associated with a host of early-life developmental indicators, including longer time to pregnancy [77], reduced embryo quality [78], lower rates of fertilization [79] and live birth [78].

### 3.4. BDE-47 Modifies Effect of Aging

Compared to age-dependent changes in sncRNA expression, we found only small effects of BDE-47 exposure on sncRNA expression. In fact, only one miRNA was differentially expressed on PND65 in response to exposure, while on PND120, six tRNA and 18 miRNA were differentially expressed. Interestingly, almost all of these sncRNA changes in response to exposure were also altered in an age-dependent manner in control animals, suggesting that exposure modifies the normal dynamics of age-dependent changes in sncRNA expression in sperm. This conclusion is also supported by the fact that unsupervised clustering of sncRNA profiles identifies distinctive age groups of control animals but not of exposed animals (Figure 2B,C). The much smaller numbers of sncRNA differentially expressed between the two age groups in exposed animals than in controls are likely the result of the following effects of exposure. First, sncRNA profiles of exposed animals undergo changes that may be interpreted as the acceleration of age-dependent changes in younger animals and their deceleration in older animals (Appendix A). We call this effect the “convergence effect” as it decreases the difference between the sncRNA profiles of animals of different ages.

Interestingly, another study from our group, using the same rat model, reported similar effects of BDE-47 exposure on age-dependent changes in sperm DNA methylation [35]. In this study, exposure had minor effects on DNA methylation when exposed and control animals of the same age were compared. Similar to sncRNA, the profiles of DNA methylation were drastically different in two age groups of control animals. In fact, 5319 age-dependent DMRs were identified in control animals and only 189 were identified in exposed rats. Similar to age-dependent changes in sncRNA, exposure produced a “convergence” effect on the sperm DNA methylation profile in exposed animals, making the methylome of young animals look older and te methylome of mature animals look younger (Figure 1).

Additionally, our data indicate that exposure increases the variance in the expression of sncRNA between biological replicates, suggesting that increased variance may, at least in part, explain why some age-dependent changes in exposed animals did not reach the significance threshold.

### 3.5. Limitations and Future Directions

Emerging evidence indicates that tRNA-derived small RNAs, including tRNA-derived fragments (tRFs), tRNA halves (tiRNAs) and rRNA-derived small RNA (rsRNA), may have significant regulatory functions in spermatozoa [66]. New bioinformatic methods have been developed in recent years to analyze the functional role of these fragments [80]. These types of small RNA are not yet well characterized and annotated in rats, making it difficult to produce a meaningful functional analysis of the observed changes in the abundance of these scnRNA. We hope that this analysis will become possible as new databases of rat small RNA are developed.

Spermatozoa are vehicles that deliver epigenetic information via several molecular mechanisms. One important mechanism that was not addressed by our studies consists of histone retention and histone modification. Future histone-focused analysis may provide a more complex understanding of the epigenome changes associated with age and environmental exposure.

Our study established the effects of age and environmental exposure on sperm sncRNA, but does not address the molecular mechanisms involved. One such potential mechanism is oxidative stress, which may mediate both effects of environmental xenobiotics and age [81]. Additional research is needed to test the involvement of oxidative stress in the epigenetic programing of sperm.

Additionally, in our study, the first time point when sperm was collected was PND65. Although the first spermatozoa appear in the caudal epididymis 2 weeks earlier, it is possible, but unlikely, that some of the spermatozoa collected on PND65 represent the first wave of spermatogenesis. Hypothetically, the first wave of spermatogenesis may not yet have a well-established process of epigenetic maturation, suggesting that some of the differences reported in our study may be explained by a comparison of spermatozoa resulting from different developmental phases of spermatogenesis maturation, rather than just from the different ages of animals. Future research is needed to analyze the changes in sncRNA over the whole reproductive lifespan.

## 4. Materials and Methods

### 4.1. Animals and Treatment

All animal protocols were conducted in accordance with the Guide for the Care and Use of Laboratory Animals and approved by the Institutional Animal Care and Use Committee at the University of Massachusetts, Amherst (approval #2013-0069, 02/26/2016). Seven-week-old Wistar rats were purchased from Charles River Laboratories (Kingston, NY, USA) on the sixth day of pregnancy, housed in a temperature- and humidity-controlled room with a 12-h light cycle and maintained at 23 ± 2 °C. All rats were fed ad libitum with a rodent chow (Prolab Isopro RMH 3000, Cat. # 5P75, LabDiet, St. Louis, MO, USA). Between pregnancy day 8 and postnatal day 21 (PND21), dams were fed daily from the tip of a pipette (0.2 µL/g body weight of vehicle; tocopherol-stripped corn oil, MP Biomedicals, Solon, OH, USA) or the same volume of a 1 mg/mL solution of BDE-47 (AccuStandard, Inc., New Haven, CT, USA; 100% purity) daily (*n =* 6 per exposure group). In the latter group, it resulted in an exposure level of 0.2 mg/kg body weight of BDE-47 per day. This method of exposure was developed to substitute oral gavage, which induces a significant stress response and may interfere with the analyzed health outcomes [82]. The litters were not culled after delivery to avoid catch-up growth that may be associated with a significant increase in nutrient availability following culling [83]. Pups were weaned on PND21. On PND65 and PND120, one male pup was randomly selected from each litter, fasted for 2 h and euthanized using cervical dislocation. Other pups were used in a different study. All euthanasia was done during morning hours, between 9 and 10 a.m. At each euthanasia, both distal cauda epididymides were collected, incised longitudinally and incubated at 37 °C for 30 min in 1 mL of sperm wash buffer (Cat. # ART1006, Origio, Denmark) to collect motile spermatozoa via the swim-up procedure. All animal experiments were performed at the University of Massachusetts, Amherst.

### 4.2. Extraction of Sperm RNA

For sperm sncRNA analysis, we used sperm collected from 6 animals per time point per exposure group. All animals were randomly selected from individual litters, one animal per litter. To remove somatic cell contamination, spermatozoa samples were loaded on top of a density gradient (40% Isolate, Irvine Scientific, Santa Ana, CA, USA) and centrifuged for 25 min at 500 g. Pelleted spermatozoa were used to extract RNA using a protocol described elsewhere [84] with modifications. In short, pelleted spermatozoa were homogenized for 5 min using the max speed setting of Disruptor Genie (Scientific Industries) and glass sand in 0.5 mL of RLT buffer with 7.5 μL β-mercaptoethanol. Homogenate was mixed with 0.5 mL QIAzol (Cat. # 79306, Qiagen, Germantown, MD, USA) and subjected to 2 more minutes of homogenization with the same settings. Homogenate was then mixed with 0.22 mL of chloroform, incubated for 3 min at room temperature and centrifugated at 12,000 rpm, 4 °C, 15 min. The aqueous phase was collected and mixed with a 1.5× volume of 96% ethanol and RNA was purified using the RNeasy Mini Kit (Cat. # 74104, Qiagen, Germantown, MD, USA) protocol with a DNase digestion step (RNase-Free DNase Set (50), Cat. # 79254, Qiagen, Germantown, MD, USA). Purified RNA was eluted in 30 μL of nuclease-free water and 1 μL of DTT and 0.5 μL of RNase Block (Cat. # 12091021, Stratagene (La Jolla, CA, USA) was added immediately after purification.

### 4.3. Preparation of sncRNA Libraries and Sequencing

Libraries of small RNA were constructed using the NEBNext Multiplex Small RNA Library Prep Set for Illumina (Cat. # E7330, New England BioLabs, Ipswich, MA, USA) as per the manufacturer’s guidelines, with a size selection of 147–160 nucleotide fragments using high-resolution gel containing 0.8% agarose, 0.4% polygalactomannan and 1.6% γ-polygalactomannan in TAE buffer. Libraries were sequenced on NextSeq 500 (Illumina, San Diego, CA, USA). RNA extraction, library preparation and sequencing were performed in the Laboratory of Evolutionary Genomics, A.N. Belozersky Institute of Physico-Chemical Biology, Moscow State University.

### 4.4. Identification of Differentially Expressed Small RNA

Read quality was checked using FastQC [85]. Adapter trimming was done using cutadapt [86]. Reads without adapters were first mapped to the UniVec [87] contaminants database to filter out non-rat sequences. In the next step, ribosomal RNAs were filtered out by aligning the remaining readings to the SILVA v132 rat ribosomal RNA database [88]. Remaining reads were aligned in the following order: miRNA > piRNA > tRNA. After each step, only unmapped reads from previous steps were used in the analysis. Reads were mapped to rat miRBase v22 for miRNA [89], piRNAdb v1.7.5 for piRNA (www.pirnadb.org, accessed on 10/04/2019) and GtRNAdb v18.1 for tRNA [90] using the bowtie aligner [91], samtools [92] for SAM/BAM data manipulation and bedtools [93] for mapped read counting. If a read was mapped to several small RNAs, it was assigned randomly to one of the sequences. The 3′CCA sequences were not removed prior to tRNA mapping. Raw read counts for microRNAs, tRNAs and piRNAs were loaded into the DESeq2 [94] R package, then processed by variance-stabilizing transformation (VST) [95], and values of differential expression were identified using default independent filtering and Cook’s cut-off parameters in the DESeq2 package [36]. VST counts were used for sample relationship visualization using PCA and a heatmap.

### 4.5. Statistical Analysis

Differences in the distribution of sncRNA subtypes were evaluated using a chi-square test. Variance in the expression values of each group were analyzed using a *t*-test. Pearson’s correlation and a *t*-test were used to compare the expression values and fold change values of significant age-dependent sncRNA between exposure groups. Fisher’s exact test was used to analyze the enrichment of piRNA targeting different genomic elements. Statistical analyses listed in this paragraph were done using Microsoft Excel 2016 (16.0.5056.1000).

### 4.6. Functional Analyses

For the enrichment analysis, based on the differential expression of miRNA, the list of significantly differentially expressed miRNAs (fold change ≥2 or ≤−2 and false discovery rate (FDR) adjusted *p* ≤ 0.05) was first uploaded into miRDB (http://mirdb.org/mining.html, accessed on 30/09/2020), an online database for miRNA target prediction [37,38], to identify the gene targets of the miRNAs. Targets with prediction scores ≥ 80 were selected for enrichment analysis, as recommended by miRDB developers. To identify molecular pathways enriched with predicted targets, Metascape [96] (https://metascape.org/gp/index.html, accessed on 10/10/2020) was used with default settings.

To analyze the functional significance of changes in piRNA expression, we first analyzed the enrichment of piRNA targeting different transposable elements (LINE, SINE, LTR and satellite) and protein-coding genes using Fisher’s exact test. Data on piRNA targets were downloaded from piRBase (http://regulatoryrna.org/database/piRNA/, accessed on 20/09/2020), which maintains a manually curated annotation of more than 4 million rat piRNA [39].

## 5. Conclusions

Our study compared sncRNA in the sperm of young pubertal and mature rats. Aging was associated with changes in the composition of sncRNAs, including age-dependent decreases in rRNA and lncRNA and increases in tRNA and miRNA. Changes in the expression of individual miRNA and piRNA were enriched for targets associated with developmental and metabolic processes. Differentially expressed piRNA were highly enriched for LTR targets. Perinatal exposure to environmentally relevant doses of BDE-47 accelerates age-dependent changes in sncRNA in younger animals, decelerates these changes in older animals and increases the variance in the expression of all sncRNA. Future research is needed to identify the mechanisms involved in age-dependent changes in sperm sncRNA profiles as well as the downstream effects on the health and development of future progeny.

## Figures and Tables

**Figure 1 ijms-21-08252-f001:**
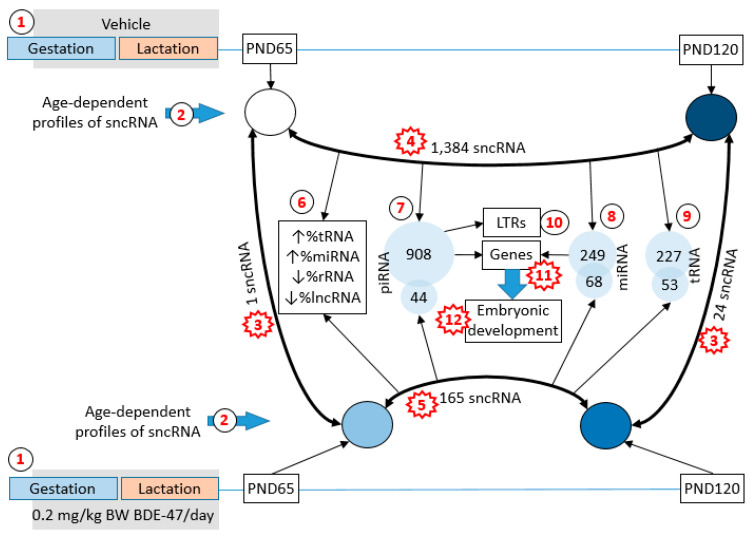
Summary of the study design and main findings. Sperm samples were collected on postnatal day (PND)65 and PND120 from rats perinatally exposed to 2,2′,4,4′-tetrabromodiphenyl ether (BDE-47) or vehicle (1). Profiles of small non-coding RNA (sncRNA) were analyzed in these samples (2). A comparison of sncRNA profiles from exposed and control animals of the same age showed only minor effects of exposure (3). Profiles of sncRNA in control animals of different ages had drastic differences (4). Age-dependent differences were attenuated in exposed animals (5) due to the “convergence” of sncRNA profiles (shown by shades, where the profile of young controls is shown in white, while profile of mature controls is dark blue, and the profiles of both age groups of exposed rats have intermediate shades). Fraction distributions of sncRNA subtypes undergo similar age-dependent changes (shown by arrows) in both exposure groups (6). Age-dependent piRNA (7), miRNA (8) and tRNA (9) were identified for control and exposed animals. Targets of piRNA were enriched for long terminal repeats (LTRs) (10) and protein-coding genes (11). Protein-coding genes—targets of piRNA and miRNA (11)—were enriched with categories related to embryonic development (12). Numbers shown in red stars indicate findings concordant with the analysis of DNA methylation in the same rat model [35]: 3—exposure to BDE-47 had a minor effect on DNA methylation if exposed and control animals of the same age were compared; 4—age-dependent differences in DNA methylation were significant in control animals; 5—these differences were attenuated by BDE-47 exposure due to the “convergence” effect, 11—gene targets of age-dependent sncRNA and genes associate with age-dependent differentially methylated regions of DNA overlapped significantly; 12—these genes were highly enriched for embryonic development.

**Figure 2 ijms-21-08252-f002:**
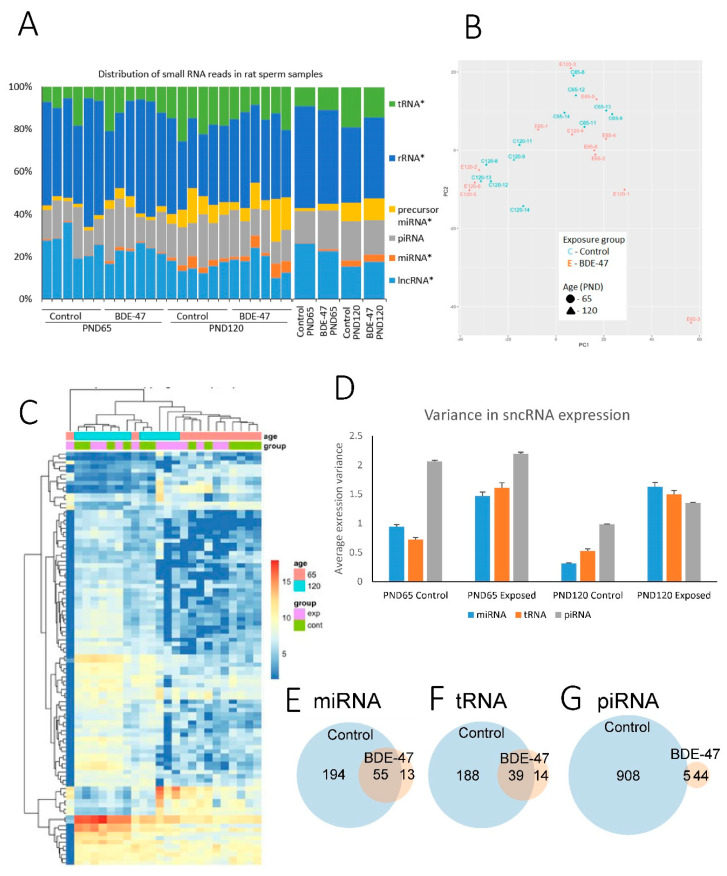
Age and BDE-47-dependent changes in profiles of sncRNA expression; (**A**)—distribution of different types of RNA fractions across biological samples, (**B**)—PCA plot of all biological samples based on miRNA expression, (**C**)—expression heatmap and hierarchical clustering of biological samples based on expression of 100 miRNA with top highest standard deviations, (**D**)—variance of sncRNA expression in age and exposure groups, (**E**–**G**)—overlap of differentially expressed miRNA (**E**), tRNA (**F**) and piRNA (**G**) in control and BDE-47-exposed animals.

**Figure 3 ijms-21-08252-f003:**
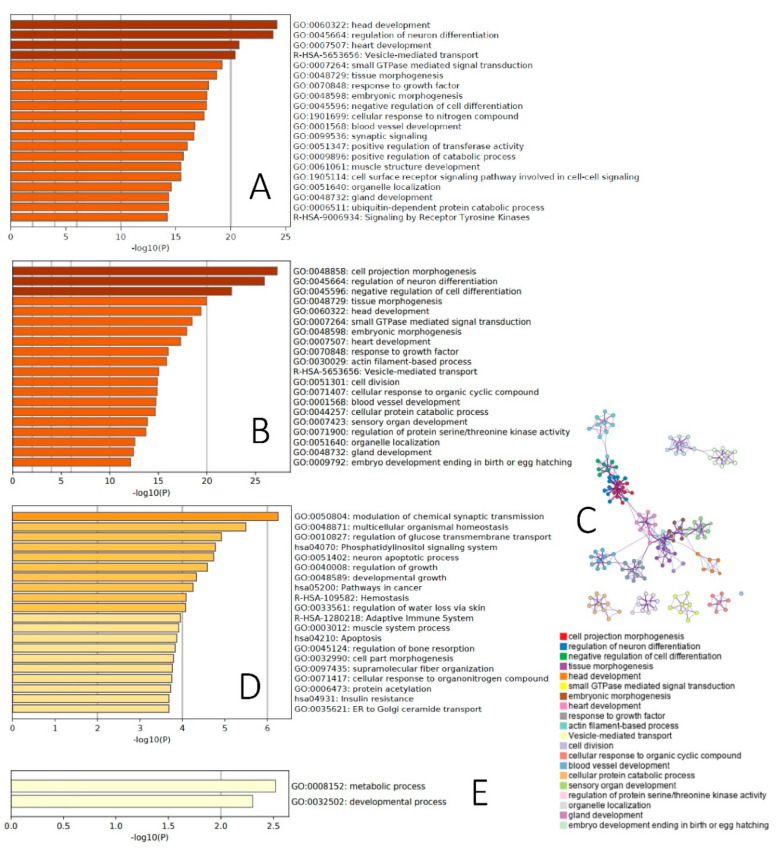
Functional enrichment of differentially expressed sncRNA: (**A**)—enrichment of biological categories with 2654 gene targets of ≥ two age-dependent miRNA differentially expressed in control animals, (**B**,**C**)—enrichment of biological categories (**B**) and network of enriched terms (**C**) for 2261 targets of age-dependent miRNA differentially expressed in both control and BDE-47-exposed animals, (**D**)—enrichment of biological categories with 688 gene targets of miRNA differentially expressed in BDE-47 animals on PND120, (**E**)—enrichment of biological categories with 42 gene targets of piRNA differentially expressed between two age groups of control animals.

**Table 1 ijms-21-08252-t001:** Number of individual miRNA, piRNA and tRNA significantly (false discovery rate (FDR) adjusted *p* ≤ 0.05) differentially expressed (twofold) in rat sperm in relation to age and exposure to BDE-47.

	Effect of Age		Effect of Exposure	
	Control	BDE-47-Exposed	PND65	PND120
miRNA *	249	68	1	18
piRNA	908	44	0	0
tRNA	227	53	0	6

* miRNA and precursor miRNA.

**Table 2 ijms-21-08252-t002:** Functional enrichment of significant age-dependent piRNA.

Targets	Identified Non-Age-Dependent piRNA	Identified Age-Dependent piRNA	Enrichment, Fold Change	*p*-Value	Direction of Change in Age-Dependent Genes
All	36,551	980	--	--	139 down 841 up
LINE	1557	33	−1.2	0.40	4 down 19 up
SINE	1648	47	1.1	0.33	6 down 41 up
LTR	521	80	5.3	<0.00001	13 down 67 up
Satellite	52	1	−1.28	1	1 up
Protein-coding	353	142	11.8	<0.00001	24 down 118 up

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
