# Peer review of "Aging Induces Profound Changes in sncRNA in Rat Sperm and These Changes Are Modified by Perinatal Exposure to Environmental Flame Retardant"

_ijms, 2020, doi:10.3390/ijms21218252_

Round 1
Reviewer 1 Report
This is a very interesting study reporting that the natural aging process has profound effects on sperm profiles of sncRNA which may also be modified by environmental exposures.
Comments
- Authors should validate their results. by choosing some representative sequences (for each group) found to have a differential expression and analyzing them for quantification of the expression.
Minor comments
- In Discussion, a figure illustrating the most interesting features would be more impressive
Author Response
This is a very interesting study reporting that the natural aging process has profound effects on sperm profiles of sncRNA which may also be modified by environmental exposures.
- We are very thankful to the reviewer for these positive comments.
Comments
- Authors should validate their results. by choosing some representative sequences (for each group) found to have a differential expression and analyzing them for quantification of the expression.
- We completely agree with the reviewer that validation of sequencing results by an alternative method should be done in this type of studies. In fact we planned to perform validation using PCR approach and have ordered all supplies needed for small RNA PCR in spring 2020. Unfortunately, due to COVID19 supply chains for scientific kits and consumables were severely affected in Russia (that is where all wet lab part of the study was done and where biological samples are stored today). We still did not receive these required kits and primers and according to supplies they cannot give even rough estimation on when the supply will be restored. Given, that situation with COVID19 is not getting better in a foreseeable future, we decided to submit this manuscript without PCR validation. We are confident however in high quality of our data, as our biological findings were supported by another project from our group, in which same group of mice was used to analyze changes in DNA methylation in sperm in relation to age and BDE-47 exposure. The results of the two studies are highly concordant. These concordances include: (1) age had significant effect on both sncRNA and DNA methylation; (2) exposure had only minor effect on both sncRNA and DNA methylation when animals of the same age were compared; (3) exposure seemingly attenuated effect of age for both sncRNA and DNA methylation by making epigenetic profiles of younger animals look older and epigenetic profiles of mature animals look younger; (4) there is highly significant overlap between gene-targets of age-dependent miRNA and piRNA and genes associated with age dependent differentially methylated regions. Thus, all major biological findings of the two studies are concordant. We included this information into the manuscript - see lines 490-496 and 520-528. We also summarized these concordances in a new figure - Fig.1.
Minor comments
- In Discussion, a figure illustrating the most interesting features would be more impressive
We have developed a new figure that summarizes the design of the study and its major findings - see Fig 1.
Reviewer 2 Report
This paper is at the top of current knowledge on iRNAs. However, the authors should be a little bit more didactic. Epigenesis is a 3-feet story with DNA Methylation, chromatin remodeling (Histones modification Arg and Lys methylation) and yes interference RNAS . Methylation and Ox stress are only quoted twice in their text….
Moreover it is clear that age 1-decreases the resistance to oxidative stress 2-EDCs induce oxidative stress (Skinner Cooney, Mannikam) leading to transgenerationnal anomalies , see also obesity and diabetes 3-Oxidative stress impairs DNA methylation. Literature is full of good papers on this specific point; Ox stress may induce abnormal de-methylation: see Especially the work in the yellow mouse and the effect of TLTR methylation. “we found only small effects of 471 BDE-47 exposure on sncRNA expression, line 471-472”This means that again DNA methylation is more important here than the miRNAs
This should be mentioned
In the mat and met the authors should not be that esoteric FDR (false discovery rate), mi RDB micro RNA Target prediction Database … VST… A lot of acronyms are not defined and reading is not that easy even if you are a biochemist. Please efforts to make it easier
In their discussion concerning the effect of the EDC and age: The effect is positive of young but not old: This is possibly related to the fact that Ox stress has reached a maximum, with no specific effect
Fig 2 diffcult to read in the text received
Author Response
This paper is at the top of current knowledge on iRNAs. However, the authors should be a little bit more didactic. Epigenesis is a 3-feet story with DNA Methylation, chromatin remodeling (Histones modification Arg and Lys methylation) and yes interference RNAS . Methylation and Ox stress are only quoted twice in their text….
- We are very thankful to the reviewer for positive comments. We also completely agree that epigenetics include 3 major mechanisms and all 3 should be addressed to have a full picture of epigenetic changes. In fact, in another study done by our group, the same model was used to analyze changes in DNA methylation associated with age and exposure to BDE-47. The results of the two studies are highly concordant. These concordances include: (1) age had significant effect on both sncRNA and DNA methylation; (2) exposure had only minor effect on both sncRNA and DNA methylation when animals of the same age were compared; (3) exposure seemingly attenuated effect of age for both sncRNA and DNA methylation by making epigenetic profiles of younger animals look older and epigenetic profiles of mature animals look younger; (4) there is highly significant overlap between gene-targets of age-dependent miRNA and piRNA and genes associated with age dependent differentially methylated regions. Thus, all major biological findings of the two studies are concordant. We included this information in the manuscript - see lines 490-496 and 520-528. We also summarized these concordances in a new figure - Fig.1. It is likely, that histone modification may have secondary significance for spermatozoa, as most of histones are replaced by protamines in spermatozoa. However, analysis of histone retention and modification may uncover an additional layer of epigentic programing in sperm. We have added that information in discussion - lines 543-547.
Moreover it is clear that age 1-decreases the resistance to oxidative stress 2-EDCs induce oxidative stress (Skinner Cooney, Mannikam) leading to transgenerationnal anomalies , see also obesity and diabetes 3-Oxidative stress impairs DNA methylation. Literature is full of good papers on this specific point; Ox stress may induce abnormal de-methylation: see Especially the work in the yellow mouse and the effect of TLTR methylation. “we found only small effects of 471 BDE-47 exposure on sncRNA expression, line 471-472”This means that again DNA methylation is more important here than the miRNAs
This should be mentioned
- We completely agree with the reviewer about potential mediating mechanism of oxidative stress in age- and chemical exposure- dependent changes in sncRNA in sperm. We have added this important future direction of research - see lines 548-552.
In the mat and met the authors should not be that esoteric FDR (false discovery rate), mi RDB micro RNA Target prediction Database … VST… A lot of acronyms are not defined and reading is not that easy even if you are a biochemist. Please efforts to make it easier
- We proofread the manuscript to ensure that all acronyms and specific slang word are explained when possible.
In their discussion concerning the effect of the EDC and age: The effect is positive of young but not old: This is possibly related to the fact that Ox stress has reached a maximum, with no specific effect
- We are very thankful to the reviewer for these thought. That is an interesting hypothesis which we may try to test in our future research. We have added information about potential importance of oxidative stress as a mechanism mediating age and exposure associated changes in sperm epigenome, - see lines 548-552.
Fig 2 diffcult to read in the text received
- We noticed that figures inserted in the text are not of highest quality. We provide figures in a separate files in our submission additionally to these inserted in the text of the manuscript.